# Boronic Acid Esters and Anhydrates as Dynamic Cross-Links in Vitrimers

**DOI:** 10.3390/polym14040842

**Published:** 2022-02-21

**Authors:** Mateusz Gosecki, Monika Gosecka

**Affiliations:** Centre of Molecular and Macromolecular Studies of the Polish Academy of Sciences, Sienkiewicza 112, 90-363 Lodz, Poland; mdybko@cbmm.lodz.pl

**Keywords:** vitrimer, boronate, boroxine, self-healing, processability

## Abstract

Growing environmental awareness imposes on polymer scientists the development of novel materials that show a longer lifetime and that can be easily recycled. These challenges were largely met by vitrimers, a new class of polymers that merges properties of thermoplastics and thermosets. This is achieved by the incorporation of dynamic covalent bonds into the polymer structure, which provides high stability at the service temperature, but enables the processing at elevated temperatures. Numerous types of dynamic covalent bonds have been utilized for the synthesis of vitrimers. Amongst them, boronic acid-based linkages, namely boronic acid esters and boroxines, are distinguished by their quick exchange kinetics and the possibility of easy application in various polymer systems, from commercial thermoplastics to low molecular weight thermosetting resins. This review covers the development of dynamic cross-links. This review is aimed at providing the state of the art in the utilization of boronic species for the synthesis of covalent adaptable networks. We mainly focus on the synthetic aspects of boronic linkages-based vitrimers construction. Finally, the challenges and future perspectives are provided.

## 1. Introduction

The development of polymer materials has been for a long time defined by the classical division to thermoplastics, thermosets [1]. Despite all the advances in the development of these classes of polymers [2,3,4], ultimately, any polymer material that falls within the classic material categories suffers from its inherent limitations, such as the lack of recyclability in the case of cross-linked materials, and poor solvent and creep resistance in the case of thermoplastics. Moreover, growing environmental awareness imposes the design of polymer materials with a longer lifetime that can be conveniently reprocessed multiple times. In addition, novel processing methods, such as 3D printing, have emerged that require new materials to take full advantage of these methods’ potential [5].

Most of these challenges can be met by introducing labile interlinks between polymer chains that can be broken under certain conditions. Under the working conditions of the material, an appropriately adjusted number of such cross-links provides the desired properties. However, upon triggered degradation/activation of cross-links, processing, and recycling of the material is possible. Various strategies for incorporating such liable bonds of different origins into polymers have been developed. These include nano- and microscale phase engineering [6], and the incorporation of various supramolecular bonds [7,8,9]. These strategies, however, are limited to a narrow range of polymers to which they can be applied, due to the set of functionalities they provide, or the high cost of implementation.

An undoubted breakthrough was the use of dynamic covalent bonds [10] for the design of polymer materials. The idea of covalent adaptable networks (CAN), and especially the degenerative approach that gave rise to vitrimers (Latin: vitrum—glass) by utilizing the advantages of both kinetics and thermodynamics of covalent bonds, proved to be the most versatile. Beginning with a malleable epoxy thermoset based on the transesterification reaction introduced by Leibler [11], the vitrimer concept has been utilized in a wide range of polymer materials, including commodity plastics [12], repairable adhesives [13], and recyclable rubber [14].

Numerous reactions were utilized for the construction of dynamic cross-links, such as carboxylic acid esters transesterification [11,15], vinylogous urethane transamination [16] urea transamidation [17,18], imine exchange reaction [19], silyl esters transesterification [20], etc. Importantly, these bonds are commonly found in polymeric materials. We refer readers to a perspective article by Winne, Leibler, and Du Prez that focuses on the mechanistic aspects of different types of dynamic cross-linking reactions used to synthesize dynamic covalent networks [21].

Similarly to carboxylic acid esters and silyl acid esters, boronic acid esters undergo a transesterification reaction. Moreover, apart from transesterification reaction, boronic esters also undergo metathesis reaction and so do boronic anhydrides—boroxines. All these reactions vary in terms of the reaction mechanism; therefore, their kinetic and thermodynamic parameters differ. Such diversity, however, combined with the simple synthesis of boronic acid-based cross-links, made boronic species a useful building block for the synthesis of covalent adaptable networks [22].

In this review, we present key examples for the utilization of boronic species for the synthesis of covalent adaptable networks. We particularly focus on the synthetic aspects in the design of boronic linkages-based vitrimers, and also on how the structure influences materials properties. We make a reservation, however, that not all examples of polymer networks quoted in this review are referred to by the authors as vitrimers. However, they are all constructed using boronate or boroxine cross-links and exploit, to a certain extent, their dynamic character to obtain the desired properties. The article starts with the discussion of general mechanistic aspects associated with vitrimers and boronic acids-based dynamic species. It is followed by state-of-the-art examples that are sorted by the type of boronic cross-links and the type of polymer matrix applied. Finally, we conclude this review with limitations and perspectives on vitrimers based on boronic derivatives.

## 2. Fundamentals of Vitrimers and Boronic Acid-Derived Dynamic Bonds

### 2.1. Boronic Esters

Boronic acids are trivalent organoboron compounds with the general formula *RB(OH)_2_*, where R is an organic substituent. The boron atom is sp^2^ hybridized and possesses a vacant p orbital. In consequence, boronic acids are mild Lewis acids and do not behave in water like typical Brønsted acids, but they ionize water and form hydronium ions by ‘indirect’ proton transfer (Equation (1)). The pK_a_ of aryl boronic acids ranges from 8.9 for phenylboronic acid to around 4.0 for 3-pyridylboronic acid [23].



(1)
$$RB{\left(OH\right)}_{2}+2\;H_{2}O\;⇄RB{\left(OH\right)}_{3}^{-}+H_{3}O^{+}$$



Boronic acids, similarly to carboxylic acids, form esters with alcohols. Acyclic boronic esters are hydrolytically unstable [24]. However, they form much stable cyclic five- and six-membered esters with 1,2- and 1,3- diols, respectively [24]. Saturated five- and six-membered cyclic boronic esters are named dioxaborolanes and dioxaborinanes, respectively. The ability of boronic acids to form stable cyclic esters with diols has attracted researchers since the late 1950s [25]. Since diol groups are the inherent part of saccharides, the binding of diols by boronic acids has been utilized for the construction of sensors [26] and separation systems for carbohydrates and glycoproteins [27,28]. Moreover, boronic esters are used as protecting groups in organic chemistry [29]. The dynamic character of boronic ester linkages, manifested by high rates of the transesterification reaction, has also attracted polymer materials scientists as it can be used for the construction of dynamically cross-linked networks, in bulk, in organic solvents, or in aqueous media to make hydrogels. Boronic acid-based hydrogels are extensively researched for biomedical applications due to their self-healing properties and stimuli-responsiveness [30,31,32,33,34,35].

When the thermodynamics and kinetics of boronic acids esterification are considered, numerous parameters must be taken into account. The boronic acid can exist in either trigonal sp^2^ or in sp^3^ hybridization, of which the tetrahedral hybridization more preferably forms an ester (Figure 1.) [36]. The apparent equilibrium constant of esterification depends on such factors as the pK_a_ of both an acid and a diol, structural factors such as conformational rigidity, stereochemical hindrance, etc. [37].

The equilibrium between trigonal and tetrahedral states plays a crucial role in the aqueous system, especially when changes in pH occur. In the case of bulk vitrimers, which we can consider as anhydrous, boron atom hybridization in ester cross-links is usually trigonal. In this state, boron atoms are Lewis acids that are well-known to interact with nitrogen donors in so-called coordinative or dative N→B bonds [38,39,40,41,42]. The equilibrium constant of complex formation via this kind of interaction can be as high as 10^6^ M^−1^ [38], which allows supramolecular constructs formation [39,40]. What is important is that the geometry of the boron atom and surrounding bonds become tetrahedral upon the complex formation (Figure 2), making boronate esters more stable [41]. Nitrogen donors such as amines, pyridines, amides, and imidazoles can be added to the systems as an additive, or they can be built-in into the structure of an acid [42] or a diol [24], which gives a lot of freedom in designing of systems that utilize these interactions. In addition, the placement of a carbonyl group adjacent to the boron center, for example, as in the case of 2-acrylamidephenylboronic acid [43], facilitates boronate ester formation due to the interaction between boron and oxygen [44]

The boronate cross-links reshuffling may be governed by two reactions, transesterification with free diol groups in the system [45] or metathesis [12]. Wulff′s study showed that the proton transfer step (Figure 3 step B) is rate-determining in the transesterification of esters [46]. Therefore, a basic group, for instance, an amine group, may substantially accelerate the transesterification reaction rate if it is properly positioned with respect to the boronate ester. Interestingly, the accelerating effect is not related to a dative bond formation. Above that, the reaction rate is determined by the structural features of an acid and diols involved in the reaction [24].

The second, actually more commonly utilized reshuffling reaction in boronate-based vitrimers, is boronate esters metathesis. The reaction was first confirmed by Rotger et al. who showed that the mixture of two boronate esters, namely, 4-ethyl-2-phenyl-1,3,2-dioxaborolane (B1-D2) and 4-methyl-2-(3,5-dimethylphenyl)-1,3,2-dioxaborolane (B2-D1), equilibrate forming the following four esters: B1-D1, B1-D2, B2-D1, and B2-D2 [12]. The kinetics of the exchange reaction followed the second-order model, assuming a single metathesis rate. The determined activation energy (E_a_) of the reaction was ~15.9 kJ/mol. In the case of six-membered ester, the reaction is slower and the E_a_ determined by Yang et al. for model small molecules was equal to 23.6 kJ/mol [47]. A much higher E_a_ (62.9 kJ/mol) has been determined for the intramolecularly stabilized diethanolamine ester with an N-B dative bond. Furthermore, it has been shown that the ester metathesis reaction accelerates in the presence of catalytic amounts of free alcohols and diols [48]. In the presence of 1 mol % excess of neopentyl glycol, E_a_ of the metathesis decreased from 23.6 to 6.9 kJ/mol [47]. The mechanism of the reaction has not been elucidated. Winne and Leibler postulated that the exchange mechanism is probably a multistep process, initiated by the formation of a zwitterionic adduct, wherein alkoxide residues can be readily exchanged between the two boron centers [21].

### 2.2. Boroxines

Boronic acids upon dehydration form six-membered cyclotrimeric anhydrides—boroxines (1,3,5,2,4,6-trioxatriborinanes, R_3_B_3_O_3_) (Figure 4). In a crystal, the structure of triphenylboroxine is close to a flat and regular hexagon, with the bond angles to 120° [49]. The phenyl rings attached o boron atoms are approximately planar to the B_3_O_3_ ring [50]. Despite cyclic structure and a sextet of electrons, it is accepted that triorganoboroxines have little aromatic character [51].

Boroxine ring formation is an entropically driven process due to the extrusion of water, while ΔH of the reaction is positive [49,52]. The common strategy for the synthesis of boroxines is so-called ligand-facilitated trimerization [53]. Boroxines form complexes, similarly to boronic acids, with Lewis bases such as amines, pyridine, etc., in 1:1 stoichiometry. It was found that ΔH profit for an adduct formation is larger in magnitude than the endothermicity of ΔH for trimerization [49]. In consequence, with the addition of a ligand, the corresponding boroxine is generated in high yield without the need for aggressive chemical or thermal dehydration.

The dynamic character of boroxines that make them a suitable linkage for dynamically cross-linked polymers is manifested in a few different ways. Initially, the reversibility of boroxine ring formation has been utilized for water-triggered healing [54]. It was demonstrated by Iovine et al. that boroxines undergo a fast exchange reaction with free acid at room temperature in solution [55]. Moreover, Ogden observed the fast reshuffling in the DMSO solution of two homo-boroxines (Figure 5). In the ^1^H NMR experiment, the peak coalescence at 85 °C was observed, and the determined E_a_ of the reaction was 81.6 kJ/mol [56], i.e., about 3–4 times more than in the case of boronic esters metathesis, but still within a reasonable range to be used in vitrimers. The mechanism of boroxines reshuffling is unknown. It may be facilitated by residual boronic acid resulting from hydrolysis with water traces [56].

### 2.3. Vitrimers

Vitrimers are defined as permanent chemical networks with dynamic covalent bonds [10] that undergo molecular rearrangements allowing the network to change its topology while maintaining a constant number of chemical bonds in the system, i.e., the network connectivity at all temperatures below degradation [21,57]. In a vitrimer, dynamic cross-links undergo a degenerative exchange reaction, i.e., the equilibrium constant of the reaction is constant at all temperatures, and so is the cross-linking density. The rate of the exchange reaction, however, increases with temperature leading to the decrease in the dynamic bonds lifetime, allowing the stress relaxation in the network and ultimately its flow. Due to the permanent nature of the network with permanent connectivity at all temperatures below degradation, vitrimers show resistance to such undesired processes as dissolution, creep, and solvent stress cracking. However, the dynamic character of the network facilitates processing and recycling despite the permanent covalent nature of the cross-links [57].

The properties of polymeric materials change with temperature, and the characteristic transition temperatures found in polymeric materials determine their operating and processing temperatures. Typically, a glass transition is observed, and in the case of semi-crystalline polymers, also melting of the crystalline phase, as described by T_g_ and T_m_, respectively. Cross-linked polymers changes from glassy solids to elastomers above their T_g_ or T_m_ for amorphous or semi-crystalline polymers, respectively. In the case of amorphous thermoplastics, viscosity sharply decreases above the T_g_, following the Williams, Landel, and Ferry (WLF) model [58] up to approximately 100 °C above the T_g_ when it can be described by Arrhenius-type temperature dependence [59,60].

The properties of vitrimers are not only determined by the phase behavior and the mobility of chains in the network, but also by the kinetics of the cross-links exchange reaction [61]. Therefore, for vitrimers to flow, enough energy is required to break the physical interactions preventing the long-range motion of polymer chains, and to overpass the activation barrier of dynamic bonds reshuffling reaction. The new characteristic transition point has been thus defined for vitrimers, which corresponds to the freezing of network topology due to the absence of exchange reaction on the timescale of the observation. In practice, it is accepted that T_v_ is the temperature at which the melt viscosity is equal to 10^12^ Pa·s [11,57]. The activation energy (E_a_) for viscous flow is commonly determined based on the stress relaxation experiment by linear fitting of plotted ln(τ) values as a function of 1000/T, where characterized relaxation time (τ) is defined as the time when the normalized stress decreased to 1/e. One must remember, however, that the E_a_ for the exchange reactions of the analogous small molecules is not the same E_a_ that describes the sensitivity of the viscosity or stress relaxation behavior to temperature changes. The latter depends not only on the activation barrier of the reaction itself but also on the structure of the network [62,63,64].

The overall thermo-mechanical behavior of a vitrimer depends on the order of individual transitions, and the one that occurs at the highest temperature determines the vitrimer flow point. Figure 1 shows general dependences of melt viscosity ln(η) versus temperature, depending on the relation between T_g_, and T_v_ for an amorphous vitrimer. If T_g_ < T_v_ then the vitrimer shows the rubbery state in between these temperatures, while above T_v_ it flows with following Arrhenius dependence with temperature, since it is mainly controlled by the cross-link exchange kinetics. In this case, high E_a_ values for vitrimers indicate the rapid decrease in viscosity upon the temperature rising, while vitrimers with low E_a_ exhibited less pronounced viscosity [62]. In the opposite case, when T_g_ > T_v_, the vitrimer is an elastic solid below T_g_. Once T_g_ is exceeded, the flow behavior follows the WLF model as the network rearrangement kinetics is diffusion-controlled and network rearrangements are dominated by segmental motions. With a further temperature increase the exchange kinetics change to an exchange reaction controlled regime, and in consequence, the flow follows the Arrhenius law (Figure 1). The complexity of the vitrimers flow and relaxation behavior, which cannot be precisely described with basic models, has given rise to extensive theoretical studies [65,66,67,68,69,70,71]. Dynamics simulations proved to be suitable to elucidate the complex structure and dynamics of vitrimers systems, as well as to predict their rheological behavior [67].

When designing a vitrimer for a particular application, whether it is a self-healable coating or a recyclable thermoset, one has to consider both the kinetics of the cross-links exchange reaction and the stiffness of the network chains. Moreover, other molecular parameters such as the average molecular weight of chains between two cross-links, the fraction of dynamic cross-links, etc. will influence the properties of the network. Therefore, only a proper combination of all these parameters will provide a desired set of properties to the vitrimer.

## 3. Boronic Acid Esters-Based Vitrimers

The history of polymer networks in which various boron acids esters played the role of cross-links dates back to the early 1940s, when silicon-based materials showing peculiar viscoelastic properties were invented [73,74]. In those days, however, the origin of the peculiar flow behavior of what we now know as “silly putty” was unknown. The explanation provided by Wright, which states that “The “friction” between hydrophilic and hydrophobic groups is known to be high. As a result of the “friction” between these groups when the materials of the present invention are subjected to a sudden but not sharp blow, (…), the opposing groups are (…), locked so that the material responds as an elastic solid and exhibits a high degree of bounce” [74], was, in a sense, close to the current knowledge of vitrimers properties at the molecular level. Over half a century later, silly putty itself, besides being a famous toy, still inspires novel research, for instance, as a base for novel advanced functional materials, as in the case of Boland’s electromechanical sensors made of grapheme-filled silly putty [75]. A renaissance of polymer networks based on boronic acids esters is strictly connected with the emergence of CANs [76] and vitrimers [11], in which the labile character of boron ester bonds can be exploited. Yet, the reversible character of dioxaborolanes had been employed for the design of self-repairing polymers almost a decade earlier by Niu et al. [77], who reported a linear copolymer of 9,9-dihexylfluorene-2,7-diboronic acid and pentaerythritol that could be hydrolytically degraded and subsequently reconstituted by removing water. 

In current studies on the boronate-based networks, two main trends can be distinguished. The first one relies on the vitrimers synthesis from low molecule resins in which one component contains cyclic boronate moiety. The cross-linking of the resins occurs as a result of the reaction between complementary reactive groups in the resin components (Figure 2), for example, thiol and vinyl groups, or isocyanate and hydroxyl groups. The properties of the synthesized vitrimer can be tailored by performing the following: changing the molecular weight of the components, for example, when telechelic oligomers are used, by changing the functionality of the components, and by adding inactive components that do not contain boronic acid ester. The second common approach is based on the cross-linking of macromolecules, including commercial polymers, with a low-molecular cross-linking agent bearing the dioxaborolane group. This is achieved either directly, without the need to modify the structure of the macromolecules, or by introducing additional functional groups into the polymer structure so that it can react with the cross-linking agent. Besides these two most common strategies, a vitrimer can be obtained in a reaction between two sets of macromolecules containing complementary groups that react with each other to form dynamic boronate cross-links.

Depending on the resin/polymer composition various cross-linkers bearing boronic acid or boronate moiety can be applied. They are either phenylboronic acid or benzene-1,4-diboronic acid derivatives (Figure 3). Esters of these acids are commonly used which react with the resin either by boronate metathesis or in another reaction, e.g., thiol addition, hydroxyl addition to isocyanate, etc., which uses additional functional groups built into the boronate cross-linker and other vitrimer components. The use of boronates rather than direct condensation is advantageous, since it excludes the need of removing water. In addition, the incorporation of highly hydrophilic diols groups into hydrophobic polymer may lead to undesired microphase separation [78].

### 3.1. Macromolecules Cross-Linking

#### 3.1.1. Polyolefins

Polyolefins are the most common polymers used in the plastics industry today [79]. Despite all the advances in polyolefin synthesis and processing that have significantly expanded their field of application, there are still challenges, such as high temperature creep compliance, that can be met by introducing dynamic covalent cross-links into the polyolefin structure. It is noteworthy that high chain mobility of polyolefins is advantageous for the vitrimer network, as it does not restrain chains diffusion and facilities cross-links reshuffling.

The first report on the polyolefin vitrimer was presented by Cromwell et al. [45], who synthesized a vicinal diol-containing polycycloocten by the copolymerization of cyclooctene, cyclooct-5-ene-1,2-diol copolymer using ring-opening metathesis polymerization. The copolymer was cross-linked with (((hexane-1,6-diylbis(methylazanediyl))bis(methylene))bis(2,1-phenylene))diboronic acid 1,2-dihydroxypropane ester in the transesterification reaction of diol moieties in the macromolecules and the cross-linker’s boronate groups. The resulting vitrimer showed 95% healing efficiency after 16 h at 50 °C and could be recycled by melt pressing at 80 °C to reform the bulk material. The crucial aspect in achieving vitrimer-like properties was the introduction of the methylamine group in the ortho position in the phenyl ring, with respect to the dioxaborolane group in the cross-linker structure. The proximity of the amine group increases the rate of transesterification reaction by up to five orders of magnitude, based on the small molecules model reaction [45]. The nitrogen atom acts as a proximal base to facilitate the proton transfer between the leaving group diol on the boronate and the protonated ammonium during transesterification [46,80].

Although Cromwell proved, in a very neat way, the applicability of dioxaborolane moieties for the synthesis of malleable, self-healing, and processable covalent polyolefin network, the proposed strategy still possesses certain drawbacks. First and foremost, it requires the synthesis of new copolymers, which may be inferior to the already synthesized polyolefins. In fact, there is a need to reduce the variety of polymers we use every day to simplify the plastic recycling stream [81]. Therefore, methods of transforming already used large-scale polymers into vitrimers to improve their properties are particularly sought after.

Röttger [12] and Caffy [82] reported methods for the transformation of high-density polyethylene (HDPE) into vitrimers via reactive processing. In the first case, HDPE was initially modified during the reactive mixing in a dicumyl peroxide-initiated radical reaction with maleimide bearing dioxaborolane group. Subsequently, the cross-linker 2,2′-(1,4-phenylene)-bis [4-methyl-1,3,2-dioxaborolane] was added to the melt to achieve the vitrimer. Caffy managed to obtain HDPE vitrimer in a single step by using bis-dioxaborolane cross-linker bearing two (2,2,6,6-tetramethylpiperidin-1-yl)oxyl (TEMPO) moieties, by which it was grafted to HDPE in the presence of dicumyl peroxide during the reactive extrusion process. In both cases, the obtained vitrimers were processable and showed similar mechanical properties to pure HPDE with lower elongation at break characteristic of cross-linked materials and improved melt strength and creep resistance. Unlike in the case of polycycloocten vitrimers, both HPDE vitrimers used dioxoborolane metathesis as the cross-links reshuffling mechanism. 

The radical processes that were applied for grafting dioxoborolane moieties to HDPE, however, also possess certain limitations. These include limited grafting efficiency, the occurrence of side reactions such as termination through recombination, which yields permanent non-dynamic cross-links, and irreversible detachment of bis-nitroxide cross-linker at temperatures higher than 210 °C. Therefore, there is still an area for process improvement. To avoid free-radical reactions in grafting dioxoborolane moieties to polyolefins, Yang et al. [83] used Diels–Alder cycloaddition click. Similarly to Röttger [12], Yang used a two-step method that includes maleimide-bearing dioxaborolane group grafting and cross-linking with 2,2′-(1,4-phenylene)-bis[4-methyl-1,3,2-dioxaborolane] via dioxaborolane metathesis. Maleimide-bearing dioxaborolane was, however, grafted to the anthracene group introduced to the ethylene and 1-tetradecane terpolymer with a 9-(but-3-en-1-yl)anthracene comonomer (Figure 4). The resulting cross-linked polyolefin elastomer showed good recyclability along with excellent resistance to heat and solvents. For instance, as low as 1.0% of the permanent deformation was observed after keeping the vitrimer under constant 5 kPa stress at 160 °C for 30 min. The activation energy of the cross-links reshuffling ranged from 18.2 to 36.5 kJ/mol, increasing with the cross-linking density of the vitrimers. The lowest determined E_a_ value is close to dioxaborolane metathesis E_a_ determined by Röttger for a model small molecule system (15.9 kJ/mol). These results indicate that in the case of boronate vitrimers based on the metathesis cross-links exchange reaction, chain mobility is the limiting factor for the network relaxation rate.

#### 3.1.2. Diene Elastomers

Unlike saturated polyolefin, for which modification is not straightforward, polydiene elastomers contain numerous moderately reactive unsaturated carbon–carbon bonds; thus, these polymers can be modified much easier than saturated polyolefins, without the need of additional macromolecules modification.

Chen et al. cross-linked commercial styrene-butadiene rubber (SBR) with a dithiol-containing boronic ester, namely 2,2′-(1,4-phenylene)-bis[4-mercaptan-1,3,2-dioxaborolane] that was directly blended with SBR on a roll mill and coupled onto the pendant vinyl groups of SBR through thermally initiated thiol-ene “click” reaction [84]. The resulting vitrimers showed healing properties, recovering approximately 80% of their strength after 24 h at 80 °C, and could be recycled by hot-pressing at 160 °C. As the cross-linking density influences the mechanical properties of the vitrimers, it also influences the activation energy of network relaxation. Similarly to Yang’s polyolefin elastomer systems [83], E_a_ increases with cross-link density, but it is even lower than that reported by Yang for polyolefin elastomer vitrimers and ranges from 7.7 to 13.8 kJ/mol. The chain mobility and the reactive group diffusion are restricted in a more densely cross-linked network. Consequently, bond reshuffling is hindered and activation energy increases.

2,2′-(1,4-phenylene)-bis[4-mercaptan-1,3,2-dioxaborolane] has been also applied as dynamic cross-linker for epoxidized natural rubber (ENR) [85]. Unlike in the case of SBR, the cross-linker was coupled with ENR in the ring-opening thiol to epoxide addition catalyzed with 4-dimethylaminopyridine (DMAP). The healing and recyclability of ENR vitrimers were on par with SBR vitrimer. To improve the mechanical performance of the ENR vitrimer, Chen et al. introduced additional dynamic interactions to the systems in the form of sacrificial bonds between Zn^2+^ ions that were provided with ZnCl_2_ and the ENR’s oxygen atoms. These bonds hold out against load at small deformations and break prior to covalent bonds (Figure 5). When a load is applied, they undergo reversible bond breakage/reformation, which provides an efficient energy-dissipating mechanism [86,87]. The incorporation of coordination bonds into the ENR vitrimer substantially improved Young’s modulus and tensile modulus of the vitrimer by approximately 400 and 250%, respectively. Importantly, self-healing capability and processability were not adversely affected. The introduction of sacrificial bonds with Zn^2+^ has also been reported for an SBR vitrimer. In this case, however, Liu [88], has modified SBR with 2-(2-benzimidazolyl)ethanethiol to incorporate imidazole moieties that form complexes with Zn^2+^ cations.

#### 3.1.3. Vinyl Thermoplastics

The advantages of introducing dynamic cross-links into the polymer structure were also introduced into other commodity plastics, such as polystyrene or polyacrylates. Multiple strategies of dioxaborolane were developed to prepare a vitrimer based on these polymers. Each of them, however, begins with the introduction of a functional comonomer bearing either a dioxaborolane moiety or one of the boronate ester-forming groups, i.e., boronic acid or a diol, which is subsequently used for cross-linking. The structures of monomers bearing boronic acid, diol groups, or dioxaborolane groups are presented in Figure 6.

The applied mechanism of the network relaxation, i.e., transesterification or metathesis, determines, to a certain extent, the vitrimer composition. To trigger transesterification, an excess of 1,2- or 1,3-diol groups must be incorporated into the polymer structure. This has been achieved using such comonomers as dopamine acrylamide, 3-allyloxy-1,2-propanediol, 2,3-dihydroxypropyl methacrylate.

Kim et al. synthesized poly(butylacrylate-co-dopamine acrylamide) [89,90] that was cross-linked with 1,4-benzenediboronic acid in the presence of triethylamine that interacts with boronic acid by N→B dative bond. In such a complex, phenylboronic acid adopts tetrahedral conformation in which it is more prone to ester formation. Moreover, tetrahedral boronate-ester bonds are highly stable to hydrolysis in these bulk polymers under humid conditions [90]. The resulting network showed water-triggered healing properties at room temperature, but it is hydrolytically stable in seawater, even up to one month. Moreover, it could be reprocessed by hot pressing at 10 MPa for 30 min at 60 °C.

Instead of using a low-molecular cross-linker, Zhang et al. [91] prepared a poly(butylacrylate)-based network by applying benzene-1,4-diboronic acid bis(2,3-dihydroxypropyl methacrylate) ester that is added at the polymerization step. Moreover, to provide an excess of diol groups 2,3-dihydroxypropyl methacrylate was applied. Since the vitrimer with dioxaborolane cross-links only exhibited poor mechanical properties, the authors incorporated ureido-pyrimidone (UPy) moieties that form dimers via strong quadruple hydrogen bonding, by applying a function comonomer, 2-(2-ureido-4[1H]-6-methylpyrimidinone)ethyl methacrylate. The addition of supramolecular cross-links significantly improved both tensile strength and Young’s modulus, up to two orders of magnitude, without loss of elongation at break, healing properties, and processability. A vitrimer sample could be remolded by hot compression at 110 °C for 30 min.

Vitrimers based on common thermoplastics, i.e., polystyrene (PS) and poly(methyl methacrylate) (PMMA) that use dioxaborolane metathesis as cross-links reshuffling mechanism were first prepared by Röttger [12]. Authors prepared styrene and methyl methacrylate copolymers with (5,6-dioxaborolane)hexyl methacrylate that were cross-linked with 2,2′-(1,4-phenylene)bis[4-methyl-1,3,2-dioxaborolane] during extrusion at 200 °C. Similarly to reported HDPE vitrimers, PS and PMMA vitrimers exhibited comparable mechanical performance to commercial thermoplastics and were less prone to creep at elevated temperatures. The determined activation energy of stress relaxation in PMMA vitrimers was 43.2 kJ mol^−1^, which is significantly higher than in the case of polyethylene copolymer vitrimers [83].

To eliminate the addition of low-molecular cross-linker, Wang et al. [92] prepared vitrimers by mixing two copolymers, from which one was equipped with 1,2-diol groups, while the other had pendant boronic acid moieties, both in the form of dioxaborolane. These moieties were incorporated using 2,3-dihydroxypropyl methacrylate and 4-vinylphenylboronic acid 1,2-propanediol ester, respectively. At an elevated temperature, boronic esters metathesis reaction occurs leading to gel formation. The separation of reactive groups between macromolecules reduces the number of defects in the polymer network. Moreover, the removal of low-molecular cross-linkers eliminates the possibility the cross-linker phase separation and self-assembly that may affect the vitrimer’s mechanical and rheological properties [93]. Both PS and PMMA vitrimers exhibited better mechanical properties and solvent resistance than neat thermoplastics without losing recyclability. Since PS macromolecules are much stiffer than, for example, SBR macromolecules, the determined energy of activation was also higher and ranged from 36 to 52 kJ/mol, depending on the cross-linking density.

To decrease the activation barrier of dioxaborolane cross-links exchange reaction in polymethacrylate networks, Yang et al. [47] incorporated a small fraction of free diols into the network structure. It has been achieved by copolymerization of butyl methacrylate, bismethacrylate cross-linking agent containing six-membered boronic esters linkage (5-ethyl-2-(4-((methacryloyloxy)methyl)phenyl)-1,3,2-dioxaborinan-5-yl) methyl methacrylate, and a diol containing methacrylic comonomer, for instance, 2,3-dihydroxypropyl methacrylate. As revealed with kinetics studies on small molecules, the presence of 1% of neopentyl glycol led to the E_a_ decrease in six-membered boronic esters from 23.6 to 6.9 kJ/mol. Interestingly, the presence of pendant diol groups in the vitrimers not only accelerated the network relaxation, which led to a decrease in vitrimer recycling temperature from 150 to 120 °C. In addition, the mechanical performance of vitrimers has been significantly improved, which most probably results from intermolecular hydrogen bonds formed with the participation of free OH groups.

The opposite result in terms of E_a_ has been shown by Wang et al. [94], who reported E_a_ of PS vitrimers in the range 120~150 kJ/mol. The authors synthesized styrene and hydroxyethyl methacrylate copolymers that were cross-linked with diisocyanate bearing nitrogen-coordinating cyclic boronic ester group. The presence of nitrogen that coordinates with the boron atom stabilizes the boronic ester [95]. In consequence, a high recycling temperature (180 °C) was required. However, it showed exceptionally high creep resistance even at 120 °C.

Below in Table 1, we summarize the examples of dynamic covalent networks based on macromolecules cross-linked with boronic acid esters moieties that were included in this review.

### 3.2. Small Molecule Resins

Several different approaches to vitrimers containing boronate esters from small molecule resins have been proposed. They are either based on direct condensation of compounds containing multiple diols and multiple boronic acid groups, or they use various addition reactions, such as urethane bonds formation, thiol-ene addition, etc., to form the network using functional molecules bearing boronic ester moieties. The summary of vitrimers synthesized from small molecule resins with boronate cross-links can be found in Table 2.

The idea of using boronates as dynamic cross-links for the construction of vitrimers started with Cash/Sumerlin works [96,97]. The authors synthesized vitrimers from the mixture of divinyl compound bearing dioxaborolane ring, and a mixture of di- and tetradiol via light-initiated thiol-ene addition. Although the mechanical performance levels of prepared networks were rather poor, they exhibited water-triggered healing properties at room temperature. Moreover, the networks kept their healing properties, even if 80% of the boronate linkages have been exchanged to non-dynamic bonds [97]. The synthesized vitrimer network relaxation was based on dioxaborolane metathesis. It was demonstrated, however, that the addition of free diols accelerated the exchange kinetics, for instance, 5% of free diols in respect to dioxaborolane rings decreased the relaxation four-fold, from 100 to 25 s.

The same strategy that combines photo-initiated thiol-ene addition with dynamic boronate rings has been, along with the adjustment of non-dynamic linkers content has been proved to be suitable for digital light processing 3D printing. Similar resin to Cash, with diallyl phthalate as a non-dynamic linker, was used by Robinson et al. [98] to fabricated complex three-dimensional geometries that exhibit dynamic character, which allows for the welding of multiple objects. In addition, an exchange with free boronic acids enables the post-fabrication under mild conditions, for example, the spatial patterning of fluorescent groups.

Thiol-ene addition, however thermally activated, has been utilized for the synthesis of soybean oil-based vitrimers. Zych et al. [99] used a resin containing soybean oil acrylate and 2,2′-(1,4-phenylene)-bis[4-mercaptan-1,3,2-dioxaborolane] for the synthesis of a dynamic network by curing at 120 °C for 24 h. Due to the relatively low relaxation E_a_ = 29 kJ/mol and low T_g_ = 10.5 °C, the vitrimer exhibited healing properties at RT and could be easily remolded by hot-pressing at 120 °C. Furthermore, the authors did not observe the sample break during tensile testing even with extension rates as fast as 500 mm/min. Instead, they were stretched into thin fibers. As Perego and Khabaz’s computational research has shown [67], the bond exchange rate has a substantial effect on the stress–strain relationships for vitrimers. Generally, the stress at the fracture point significantly decreases for vitrimers with a slower exchange rate, due to an insufficient number of successful rearrangements in the timescale of the external deformation. Therefore, the network does not have enough time to carry out as many rearrangements as it performs in an equilibrium condition. The outstanding healing properties of these soybean oil-based vitrimers combined with good metal adhesion make them a promising material for self-healing coatings.

Zeng et al. demonstrated the possibility of using 2,2′-(1,4-phenylene)-bis[4-mercaptan-1,3,2-dioxaborolane] as a curing agent for commercially available epoxy resins. The authors prepared vitrimers from *o*-cresol formaldehyde epoxy resin in 4-dimethylaminopyridine-catalyzed thiol-epoxy ring-opening addition reaction [100]. The resulting network showed a healing efficiency of above 95% at 160 °C after 12 h. In addition, it could be easily reshaped at 150 °C like a thermoplastic polymer and could be efficiently reprocessed by hot-pressing at 200 °C.

Boronic esters linkages with much slower exchange kinetics have been applied by Zhang et al. [101] for the synthesis of polyurethane and poly(urea-urethane) vitrimers. The resin consisted of hexamethylene diisocyanate trimer and either 4-(hydroxymethyl) phenylboronic acid triethanolamine ester or 4-(hydroxymethyl) phenylboronic acid diethanolamine ester for the synthesis of polyurethane (PU) and poly(urea-urethane) (PUU) vitrimers, respectively. The slower kinetics is provided by the internal boron−nitrogen coordination. The determined E_a_ of metathesis for model small molecules bearing internally coordinated boron atoms was 62.9 kJ/mol. Upon incorporation of this moiety into the network structure, E_a_ increased to 138.4 and 130.0 kJ/mol, for polyurethane and poly(urea-urethane), respectively. Both PU and PUU showed solvent resistance and mechanical properties like general thermosets but were recyclable at high temperatures, over 150 °C. In addition, vitrimers were hydrolytically stable as a result of the highly hydrophobic structure of the nitrogen-coordinated boronic ester linkages.

Vitrimers formed from small molecule resin that is being cured directly by condensation of diols with boronic acid were reported by Lei et al. [102], who usedsiloxane framework for the synthesis di- and tetradiols in the reaction thioglycerol with divinyltetramethyldisiloxane or 1,3,5,7-tetravinyl-1,3,5,7-tetramethylcyclotetrasiloxane, respectively. The mixtures of polydiols were subsequently cured with 1,4-benzenediboronic acid to form vitrimers. Depending on the di- to tetradiol ratio that determines the cross-linking density, the mechanical performance of the cross-linked polymer varied from very stretchable, like a cross-linked elastomer, to stiff and brittle, characteristic for thermosets. The apparent activation energy of relaxation was approximately 90 kJ/mol and did not differ significantly between vitrimers. The composition has also got a significant influence on the reprocessability of the vitrimer. Elastomer-like vitrimers could be recycled at temperatures as low as 25 °C, whereas stiff networks required 120 °C to be reshaped and to fully recover their mechanical performance.

## 4. Boroxine-Based Vitrimers

Relatively recently, researchers turned their attention toward boroxines as dynamic covalent cross-linking moieties for the dynamic polymer networks. Due to the properties of boroxines, the ease of synthesis and their geometry, they are extensively used in the synthesis of covalent organic frameworks (COF) [103,104], which are, however, not a subject of this review.

Boroxine moiety was initially used as a building block of polymer materials due to its other properties, rather than its dynamic character. Yang et al. [105] utilized boroxine’s high Li^+^ transference number, good electrochemical and thermal stability for the synthesis of polymer electrolyte based on boroxine-coupled poly(ethylene glycol). The authors achieved it in the reaction of PEG and B_2_O_3_, which resulted in PEG chains directly connected to boron atoms in boroxine rings. Li et al. [106] prepared light-emitting devices using easily processable α,ω-bis(dihydroxyboranyl)oligofluorenes which were subsequently cured by condensation of boronic acid moieties making emitting layers more stable. Interestingly, the authors of the abovementioned systems did not explore the mechanical properties of their systems provided with boroxine cross-links. It was not until 2016 when Lai and coauthors [54] reported the first healable polysiloxane network with boroxine rings as cross-links.

The common strategy is to use short telechelic diacids that form the network upon dehydration (Figure 7A). Alternatively, phenylboronic acid that has got an additional reactive group, is converted into an anhydride, which subsequently reacts with the resin’s second component that possesses complementary reactive groups to those in boroxine (Figure 7B). If the second component is a telechelic molecule, the structure of the network is the same as in the case of a single-component resin. In this case, however, the reaction responsible for the network formation is the reaction between resin components, not a boroxine ring formation. Such a situation may be advantageous for example if the curing must be carried out under conditions (time, temperature) that are not favorable for boroxines formation. Imine, amine, and amide bonds are commonly used to join boroxines with the resin. Unlike esters, anhydrides require only one type of reactive group to form the network i.e., an acid. Therefore, the composition of the network-forming resin may be less complex than in the case of a boronic ester-based network it may even be a single component resin. Moreover, in the case of boroxine-based networks, the functionality of boronic acid moiety is equal to 3, thus the number of boronic acid moieties required to form gel is lower than in the case of esters.

As mentioned in Section 2.2, the desired reprocessability and healing properties of boroxine-based networks are provided by two reactions (processes), i.e., boroxine exchange reaction or reversible boroxine acids condensation. The controlled shift in the boronic acid/boroxine equilibrium is utilized for the triggering network’s healing properties. No matter which of the above-mentioned strategies of the network synthesis is applied, the obtained network possesses little free boronic acid groups. For that reason, boroxine-based networks most often require activation to become fully healable. This is achieved by exposition of the material to water (either directly or by increasing environment humidity). The surface boroxine cross-links are hydrolyzed and can easily recombine when put together and dried. In addition, the hydrolysis of cross-links increased chain mobility on the surface facilitating healing. The exposition to water, however, is not mandatory, and in certain cases, for instance, when a polymer’s high hydrophobicity prevents penetration of the network with water molecules [107], it is redundant. As shown by Ogden [56] and Yang [108], the boroxine exchange reaction is fast enough that it enables healing and recycling at moderate temperature, without the need for cross-links hydrolysis.

To achieve healing properties, both proper cross-links exchange dynamics and network elasticity are required [109]. Therefore, for the synthesis of boroxine vitrimers low T_g_ polymers/oligomers such as PDMS (−124 °C), and polypropylene oxide (−68 °C), are usually used. Below we highlight recent advances in boroxine-based dynamic networks. They have been organized based on polymer applied for the network synthesis. The overview of included works can be found in Table 3.

### 4.1. Polysiloxane-Based Systems

As mentioned earlier, Lai’s boroxine-containing poly(dimethylsiloxane) (PDMS) network was the first reported boroxine-based dynamic covalent network/vitrimer [54]. The network was obtained in the amidation reaction of oligomeric, telechelic bis(3-aminopropyl) PDMS’s amine groups with acid chloride groups of 4,4′,4″-(1,3,5,2,4,6-trioxatriborinane-2,4,6-triyl)tribenzoyl. PDMS upon cross-linking became stiff with a tensile modulus of 182 MPa and strong, as it was able to bear a load of more than 450 times of its weight. The surface after wetting, however, becomes sticky and soft, enabling the polymer healing ability. It regained 95% of its strength when healed for 5 h at 70 °C. Such a temperature was crucial in healing processes as T_g_ of the network was determined at 65 °C. Nevertheless, the possibility of network remolding was not reported. The authors used the polymer as a matrix of a healable semi-transparent conductive composite with silver nano-wires (Ag-NW). The conducive material was pressed to water pre-treated polymer surface, which provided good adhesion between Ag-NW and support, and subsequently dried to permanently fix the conductor in the polymer outer layer. The sheet resistance of the composites was 15 Ω/sq.

4,4′,4″-(1,3,5,2,4,6-trioxatriborinane-2,4,6-triyl)tribenzoyl chloride and telechelic bis(3-aminopropyl) were also applied by Yu [111]. The authors, however, modified one of the PDMS amine groups with UPy (Figure 8), a well-known supramolecular motif, in the reaction with 2(6-isocyanatohexylaminocarbonyl-amino)-6-methyl-4[1H]-pyrimidinone. Such a combination of dynamic covalent and supramolecular bonds resulted in a softer material than the reported by Lai, with T_g_ = −13 °C and Young’s modulus = 130.46 MPa. Due to its lower T_g_, it reached 98% healing efficiency after 6 h at 40 °C. The introduced UPy motifs worked both as supramolecular cross-links between macromolecules and as binding sites for the UPy-modified graphene aerogel that was applied as reinforcing filler. The addition of 5.2 wt % of the filler significantly increased the composite performance with an over five-fold increase in tensile strength over a three-fold increase in Young’s modulus. The high strength of the composite arises from the H-bonding cross-links between the modified filler and the polymer at the interface. H-bonding cross-linking favors the structural self-healing of the composite that exhibited 79% healing efficiency after treatment at 40 °C for 6 h.

A different approach to the design of polysiloxane vitrimers was presented by Liang et al. [112], who applied PDMS (M_n_ 11 000) with pendant aminopropyl groups (up to 10 mol %). The authors have also used 4,4′,4″-(1,3,5,2,4,6-trioxatriborinane-2,4,6-triyl)tribenzoyl chloride as a boroxine cross-linker. Such a design, in contrast to the abovementioned examples, in which short telechelic PDMS was used, led to soft elastomer with maximum elongation at break up to 307%, and Young’s modulus up to 11.18 MPa. Very low T_g_ = −102 °C of the cross-linked polymer enabled booth healing and processing at room temperature. The cut pieces of the elastomer were remolded under a pressure of 0.2 MPa for 5 h at room temperature. The polymer reached 91% healing efficiency at room temperature after 48 h. Although the possibility of polymer remolding at such low temperatures may not always be useful, as low creep resistance can be expected at room temperature, this example demonstrates the importance of the combination of the network chain’s mobility and cross-links exchange reaction kinetics in the design of vitrimers.

### 4.2. Polyether-Based Systems

Polypropylene glycol (PPG) became the polymer of choice to numerous studies on boroxines based networks. Its chain elasticity, along with a high level of molecular homogeneousness and control over molecular weight provided by anionic ring-opening polymerization of propylene oxide, make PPG a great framework for the synthesis of vitrimers. Although PPG stays for glycol, actually amine-terminated PPG is commonly used for the synthesis, since the amine group is much more reactive than the hydroxyl group. In addition, it facilitates the introduction of a nitrogen atom in the vicinity of the boron atom enabling N→B dative bond formation. Since PPG has only two inherent reactive groups, at the chain ends, the resulting network is to be relatively uniform throughout its volume, yet such network defects as loops may occur. This gives the possibility to easily tailor the network properties only with the molecular weight of applied glycol.

The length of applied PPG on the properties of the boroxine cross-linked network was demonstrated by Delpierre et al. [113]. The authors prepared networks from PPG of 400 and 2000 in the reaction of amine-terminated PPG and 2,2′,2″-(1,3,5,2,4,6-trioxatriborinane-2,4,6-triyl)tribenzaldehyde with the formation of an imine bond between these molecules. It has been shown that the dative bond within imino-boroxine species makes the product more robust against hydrolysis due to the enhanced thermodynamic stability of the cross-links [119]. The longer PPG molecules provided high elasticity to the network, but also made it easily permeable to water molecules, even to ambient moisture. On the one hand, this allowed the network to self-heal easily, but on the other hand, the network was losing its integrity due to the hydrolysis of boroxine bonds with water from ambient humidity. By using shorter PPG, the polymer became more hydrophobic, the cross-linking density increased, and the mobility of the chains decreased, giving less access for water molecules. In consequence, the network was stable at ambient humidity conditions but required a longer time for healing. Nevertheless, the simple PPG 400-based boroxine network showed poor mechanical performance with Young’s modulus equal to 61 MPa and tensile strength equal to 3.9 MPa.

The extreme case of going down with the molecular weight of chains between boroxine cross-links was shown by Ogden ad Guan [56]. The authors prepared the network from a single-component resin made of diethylene glycol-bind bisphenylboronic acid. Since there were no built-in nitrogen atoms in the resin structure that could form N→B dative bonds, the resin was cross-linked in the presence of pyridine which was the catalyst for the boroxine ring formation. The resulting vitrimer exhibited high mechanical strength with Young’s modulus of 768 MPa and tensile strength of 32.9 MPa. Moreover, when pyridine was modified with a long aliphatic chain in position 4, it also acted as a plasticizer. For instance, when 4-undecylpyridine was used, the obtained polymer was more ductile and tougher (toughness increased from 0.97 to 1.98 MJ/m^2^). Such a short linker between boroxine rings made the network relatively stable under moderate humidity conditions, without losing processability, as it could be remolded by hot pressing at 80 °C. The activation energy of the relaxation vitrimer process was calculated to be 79.5 kJ/mol.

Bao et al. showed that the properties of the PPG-boroxine network can be also tailored by using PPG of different topologies [115]. Authors reported the synthesized a series of networks from the mixture of linear and tri-arm PPGs, both of similar molecular weight M_n_ ~ 400, with phenylboronic acid moieties at their ends. These were synthesized in the reaction of amine-terminated linear and tri-arm PPG with 2-formylphenylboronic and a subsequent reduction in imine bond with NaBH_4_. The secondary methylamine group in the ortho position in respect to B(OH)_2_ group facilitates boroxine ring formation similarly to the imine group. The amine linkage is, however, neither dynamic nor hydrolyzable. By changing the ratio of bifunctional to trifunctional PPG macromolecules from 1/0 to 3/2, the researchers changed the character of the cross-linked polymer from soft rubber-like with Young’s modulus 63.9 MPa, tensile strength 5.95 MPa and 376% strain at break, to much stiffer, with Young’s modulus = 331,7 MPa, higher tensile strength 22.85 MPa, but lower strain at break 23%. The obtained materials were found to be recyclable at 60 °C without loss of mechanical properties after three recycling cycles.

To obtain a healable polymer network showing good mechanical performance, instead of PPG, Delpierre et al. applied NH_2_-terminated polyhydroxyurethane synthesized from poly(propylene glycol) dicyclocarbonate (PPGBC) M_n_ ~ 640 g/mol, and 1,8-diaminotriethyleneglycol [116]. The incorporation of interchain H-bonds drastically changed the characteristics of the cross-linked polymer, making it stiff instead of rubber-like in the case of a PPG-based network. The Young’s modulus increased an order of magnitude to 551 MPa, tensile strength increased to 11 MPa, while strain at break decreased only to 3%. The higher stiffness imposes a higher temperature at which the polymer shows self-healing properties (70 °C). It has been shown that despite its stiffness, the polymer can be processed by fused deposition modeling (FDM) 3D printing like thermoplastic material, making it a malleable thermoset. Importantly, the author highlighted the cooperative effect of imine-boroxine bonds (Figure 9). If solely boroxine cross-links have been used, the obtained network was recyclable but extremely brittle. Oppositely, when boroxine rings were changed to phenyl rings, and imine bonds were the only dynamic bonds in the network, a two-fold decrease in Young modulus and an increase in strain at break up to approximately 100% has been observed. The same combination of imine-boroxine bonds has been also utilized by Yang et al. for the synthesis of polybutadiene vitrimer [107]. Authors used polybutadiene with a high 1,2-addition degree (90%), which in the first step was grafted with 2-aminoethanethiol in a light-initiated thiol-ene addition. Subsequently, amine groups attached to the macromolecules were reacted with aldehyde groups of 2-formylphenylboronic acid to form imine bonds. The modified polybutadiene was cured by dehydration that led to boroxine cross-links formation. Interestingly, as the cross-linking degree of vitrimers increased from 6 to 12%, E_a_ of relaxation decreased from 72.53 kJ/mol to 57.95 kJ/mol. It is explained by the increased likelihood of effective collision among the polymer chains at higher concentrations of imine-coordinated boroxine cross-links in the network.

Even better mechanical performance has been achieved by the combination of epoxide resin with boroxine imine-boroxine cross-links. Yuan et al. synthesized the network from diglycidyl ether of bisphenol A, amine-terminated PPG (M_n_ = 400) and 2,2′,2″-(1,3,5,2,4,6-trioxatriborinane-2,4,6-triyl)tribenzaldehyde [117]. The Young modulus of the network reached 1 GPa, with tensile strength = 37 MPa and a much higher strain at break (14.5%) than in the case of the urethane-boroxine network [116]. The combination of permanent and dynamic cross-links gave very good mechanical properties, which are on par with commercially available epoxy resins, with reasonable healing properties, i.e., 80% water-triggered healing efficiency after 12 h at 80 °C. The authors claimed that the prepared epoxy resin may have potential application as healable adhesives due to its good adhesion to aluminum revealed in a single-lap shear test.

Besides the direct incorporation of additional cross-links into the network structure at the synthesis stage, the properties of the PPG-boroxine network can be improved by the addition of another polymeric component that can interact with the network. Bao et al. [114] showed that the addition of high molecular weight poly(acrylic acid) (PAA) of M_n_ = 450,000, significantly improved the mechanical properties of the PPG-boroxine network that was synthesized through the trimerization of ortho-aminomethyl-phenylboronic acid groups at the terminals of PPG chains, with a subsequent reduction in imine bonds to secondary amines. PAA interacts with secondary amine groups in the network structure via hydrogen bonds that act as additional cross-links. The composite network with 40% of PAA reached tensile strength of 12.7 MPa with 182% strain at break and Young’s modulus of 112.45 MPa. Moreover, the composite was remoldable at 50 °C under a pressure of 0.4 kPa and exhibited nearly 90% water-triggered healing efficiency after 18 h at room temperature.

The boroxine-cross-linked thermoset showing extremely high mechanical performance has been shown by Lu et al. [118]. The authors synthesized a series of telechelic poly(aryl ether ketone)s (PAEK), with M_n_ ranging from 5100 to 17,600, through the polycondensation of bisphenol A (BPA) and 4,4′-difluorobenzophenone (DFBP) that were end-capped with 4-hydroxyphenylboronic acid. After curing via boroxine rings formation, depending on the molecular weight of PEAK used, thermosets exhibited tensile strength from 60.5 to 97.8 MPa and Young’s modulus from 1.59 to 4.10 GPa. Such high mechanical performance resulted from both π–π interactions between aromatic moieties in polymer chains and boroxine cross-links. The formation of the PEAK-boroxine network was intended to overcome difficulties related to the synthesis and processing of high-molecular-weight PEAK. The authors showed that these PEAK networks can be processed multiple times without loss of mechanical performance using a solvent-assisted (dioxane/ethanol) approach, in which ethanol breaks boroxine rings.

### 4.3. Vinyl Monomers Based Systems

The vinyl polymers are omnipresent. Therefore, the possibility of the introduction of boronic acid functionality into their structure is highly desired as it provides new, desired properties to the commodity polymers. Monomers such as 4-vinylphenylboronic acid or 3-acrylamidophenylboronic acid are commercially available thus the introduction of a boronic acid moiety at the polymerization stage is possible. Moreover, the controlled radical polymerization toolbox allows precise tailoring of macromolecules structure and in consequence material’s properties. In addition, Li et al. demonstrated that boronic monomers can be used either in the form of acid or as boroxine without losing control over polymer structure, only by the addition of a small quantity of water to control the acid–boroxine equilibrium during the polymerization [120]

The boroxine-cross-linked network made from a vinyl copolymer was presented by Yang et al. [108]. The authors synthesized a random copolymer of 4-vinylphenylboronic acid (4-VPBA) and octadecanoxy poly(ethylene glycol methacrylate), (PEGMA) in a free-radical process. Upon drying and boronic acid moieties trimerization, the copolymer formed the network that exhibited 99% water-triggered healing efficiency after 24 h at 70 °C. The mechanical properties of the network depended on the ratio of the comonomers. With the molar ratio of 4-VPBA to PEGMA changing from 2:1 to 5:1, the tensile strength of the thermosets increased from 9.3 to 27.5 MPa, Young’s modulus increased from 58 to 255 MPa, while strain at break decreased from 175 to 21%.

## 5. Conclusions and Perspectives

The introduction of covalent adaptable networks, and especially vitrimers, is undoubtedly one of the most important breakthroughs in the field of polymer materials in recent years. Numerous studies on the use of boronates and boroxines as dynamic covalent bonds have proven that these linkages are useful and versatile in terms of the properties they provide and the ease of introducing them into various types of polymers. Most importantly, it has been shown that they can be incorporated into large-scale polymers such as HDPE, PS, SBR, using a reactive processing approach which is an important step toward industrial applications.

Apart from transforming commodity polymers (HDPE, PS, SBR, etc.,) into vitrimers to give them such properties as recyclability, creep resistance, etc., most studies report the synthesis of stiff vitrimers showing mechanical properties characteristic to the thermosets. Nonetheless, there are still few application-oriented reports on the use of such thermosets. However, taking into account that those vitrimers often exhibit good adhesion and healing properties at ambient temperature, the development of self-healable coatings and adhesives based on boronate/boroxine resins is anticipated. Furthermore, it has been shown that vitrimers based on both boronate and boroxine cross-links can be used as matrices in composites filled with various types of fillers, including glass fibers [94], carbon nanotubes [121], graphene [111]. This area of application of vitrimers is particularly important due to the possibility of extending the lifetime and enabling recycling of composite materials, thanks to the use of vitrimer matrices.

Yet, one must be aware of the limitations that come with boronic acids derivatives. First of all, both boronates and boroxines are susceptible to hydrolysis, since their formation is an equilibrium process. While the on-demand shift in equilibrium by exposure to water was used to facilitate the healing process, the uncontrolled water absorption by the vitrimer will most likely lead to a deterioration of its mechanical properties, due to the degradation of the cross-links. There are known strategies to at least partially overcome this problem. Nonetheless, hydrolysis might still be an issue during long-term usage under high humidity. Therefore, one has to take it into account when designing a vitrimer system, especially if the polymer matrix may absorb water.

Another aspect that requires consideration is relatively low E_a_ for boronates and boroxines reshuffling. Low E_a_ correlates with low T_v_; thus, most often the cross-links exchange reaction will not be a limiting factor for the network relaxation, but the rigidity of the polymer framework. For that reason, boronic acid derivatives-based vitrimers may not be suitable for high-temperature applications.

Despite all the developments in the field, there is still a need for basic studies. There is still a lack of knowledge about the mechanisms behind boronates and boroxines metathesis. Only when we fully understand these reactions will we be able to fully use their potential for the synthesis of vitrimers with the desired properties.

## Data Availability

Not applicable.

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
