# Peer review of "Boronic Acid Esters and Anhydrates as Dynamic Cross-Links in Vitrimers"

_polymers, 2022, doi:10.3390/polym14040842_

Round 1
Reviewer 1 Report
This is the first review of vitrimers based on boronic acid esters and boroxines. It is clearly written and provides an interesting and comprehensive analysis of the different chemistries used in their synthesis and the properties of the resulting vitrimers. Following a general introduction, an overview of dynamic networks based on boronic acid esters (Tables 1 and 2), and boroxine crosslinks (Table 3) is presented. A final section of conclusions and perspectives provides a balance of their excellent potential for specific applications with their intrinsic limitations (water resistance). I recommend publication of this review with a minor revision to consider the following comment.
I gave a look at a set of recent reviews that analyze the variety of chemistries employed in the synthesis of vitrimers. All of them include a section presenting selected articles related to the chemistry based on boronic acid derivatives. While several of these references are commented in the present review, there are others that were not included. Possibly some of them are outside the specific focus of the review. Please, give a look to the following list and include those references that could contribute to your review:
J. N. Cambre, B. S. Sumerlin, Polymer 52 (2011) 4631e4643.
L. He, D. E. Fullenkamp, J. G. Rivera, P. B. Messersmith, Chem.Commun. 2011, 47, 7497-7499.
Y. Guan, Y. Zhang, Chem. Soc. Rev. 2013, 42, 8106-8121.
W. L. Brooks, B. S. Sumerlin, Chem. Rev. 2015, 116, 1375-1397.
T. C. Tseng, F. Y. Hsieh, P. Theato, Y. Wei, S. H. Hsu, Biomaterials 2017, 133, 20
S. H. Hong, S. Kim, J. P. Park, M. Shin, K. Kim, J. H. Ryu, H. Lee, Biomacromolecules 2018.
M. E. Smithmyer, C. C. Deng, S. E. Cassel, P. J. LeValley, B. S. Sumerlin, A. M. Kloxin, ACS Macro Lett. 2018, 7, 1105
W. Ge, S. Cao, F. Shen,Y. Wang, J. Ren, X. Wang, Carbohydr. Polym. 2019, 224, 115147.
Breuillac A, Kassalias A, Nicolay R., Macromolecules 2019; 52:7102–13
Yanning Zeng, Shuxin Liu, Xinmeng Xu, Yang Chen, Faai Zhang, Polymer 211 (2020) 123116
Besides, in the second line of the introduction complete the reference to odnosnik encyklopedia.
Author Response
Thank you for your remarks.
We have added references we found relevant (Ref 33, 35, 48, 106) to our article.
Reviewer 2 Report
This manuscript presents a review of the development of dynamic cross-links, and a special focus is put on the synthetic aspects in the design of boronic linkages-based vitrimers and on how does the structure influences materials properties. Two main types of vitrimers are introduced: the boronic acid esters-based vitrimers and the boroxine-based vitrimers.
Overall speaking, the manuscript gives a good review on the state of the art in the utilization of boronic species for the synthesis of covalent adaptable networks. It provides some new insights into the study of vitrimers. Therefore, the reviewer can recommend it for publication. There are three main comments, which should be considered:
- Some other aspects of vitrimers should also be introduced in this review paper, such as the mechanical properties, and their dependence on the dynamic cross-links.
- There are some constitutive models for the properties of vitrimers, such as some models for the self-healing and some models for the dynamic bonds. The authors should give some introduction on these models.
- Can the authors give more details on the comparation of these different types of vitrimers?
Author Response
Thank you for your comments.
We have complied to them as follows.
References to computational works on vitrimers have been added with a proper comment in section 2.3.
As indicated in the introduction, our article focuses mainly on the synthetic aspects of boronic acid-based vitrimers, and in our opinion extending the scope of the article to the physics of vitrimers would make it less clear for readers. We have added a comment regarding the influence of cross-links dynamics on the mechanical properties in section 3.2. Moreover, the properties of reported systems that were included in our review differ significantly, and depend mostly on the matrix applied. Therefore, making direct comparisons between reported vitrimers is hardly possible. In fact at there is still lack of proper systematic research on the influence of various network parameters on the properties of the vitrimers.
Reviewer 3 Report
This manuscript reviews “Boronic acid esters and anhydrates as dynamic cross-links in vitrimers”. A lot of examples are used to demonstrate the factors influencing the formation of boronic esters and boroxines and their application in the vitrimers, and the authors have pointed out that hydrolysis and the relatively low Ea values of borates and boroxines must be considered when designing polymer networks. Generally speaking, I suggest that this article can be published, but the paper needs minor revision before acceptance for publication. My detailed comments are as follows:
- Some pictures need to be updated to be clearer, such as scheme 1 and scheme 3. And the authors need to add diagrams for better description in 3.1.1.
- Citation of recent literature is suggested in the mechanistic explanation of boronic esters and boroxines. And examples can be added to illustrate the protection of catechol by boronic esters (Macromolecular Rapid Communications, DOI: 1002/marc.202100916).
- A discussion on the effect of Ea value on vitrimers performance can be added in 2.3.
- The title of 3.1.2 is proposed to be changed to the polymer name for consistency.
- Some typos need to be checked, such Page 4 “…it has been shown that the ester exchange reaction...” should be corrected to “metathesis”, Page 5 “with Lewis acids such as amines” should be corrected to “Lewis bases”
- The manuscript contains some mistakes. Such as “Young’s modulus” in Table.
- The authors could consider adding the following review articles into references which would again increase the interest to general smart polymer readers: Chemical Society Reviews, 2021, 50, 8319-8343; Advanced Materials‚ 2021‚33, 2104355; Advanced Functional Materials‚ 2021‚ 31, 2103391.
Author Response
Thank you for your comments.
We have complied to them as follows.
The quality of Schemese 1 and 3 have been improved and we have added one additional figure to the section 3.1.1.
Citation has been updated with References 34, 41, 42.
The title of 3.1.2 has been changed to: Diene elastomers.
The indicated errors have been corrected.
Comments on the influence of Ea vitrimers properties have been added to section 2.3.